# Oxygen-Vacancy-Induced Enhancement of BiVO_4_ Bifunctional Photoelectrochemical Activity for Overall Water Splitting

**DOI:** 10.3390/nano14151270

**Published:** 2024-07-29

**Authors:** Huailiang Fu, Qingxiu Qi, Yushu Li, Jing Pan, Chonggui Zhong

**Affiliations:** 1School of Physics and Technology, Nantong University, Nantong 226019, China; qiqingxiu131@163.com (Q.Q.); 2202310031@stmail.ntu.edu.cn (Y.L.); 2Research Center for Quantum Physics and Materials, Nantong University, Nantong 226019, China; 3College of Physics Science and Technology, Yangzhou University, Yangzhou 225002, China; jp@yzu.edu.cn

**Keywords:** overall water splitting, BiVO_4_ (110) facet, bifunctional photoelectrochemical activity, oxygen vacancy

## Abstract

Hydrogen generation via photoelectrochemical (PEC) overall water splitting is an attractive means of renewable energy production so developing and designing the cost-effective and high-activity bifunctional PEC catalysts both for the hydrogen evolution reaction (HER) and the oxygen evolution reaction (OER) has been focused on. Based on first-principles calculations, we propose a feasible strategy to enhance either HER or OER performance in the monoclinic exposed BiVO_4_ (110) facet by the introduction of oxygen vacancies (O_vacs_). Our results show that oxygen vacancies induce charge rearrangements, which enhances charge transfer between active sites and adatoms. Furthermore, the incorporation of oxygen vacancies reduces the work function of the system, which makes charge transfer from the inner to the surface more easily; thus, the charges possess stronger redox capacity. As a result, the O_vac_ reduces both the hydrogen adsorption-free energy (Δ*G*_H*_) for the HER and the overpotential for the OER, facilitating the PEC activity of overall water splitting. The findings provide not only a method to develop bifunctional PEC catalysts based on BiVO_4_ but also insight into the mechanism of enhanced catalytic performance.

## 1. Introduction

Driven by growing concerns about environmental pollution and the energy crisis, developing renewable “green” energy has become one of the most concerning scientific topics [1,2]. Photoelectrochemical (PEC) water splitting is one of the most feasible approaches to generating hydrogen and oxygen using solar energy. The photogenerated charge carriers involve a water redox reaction at the catalyst surface [3,4,5]. There are two electrolysis steps in the water-splitting devices, the hydrogen evolution reaction (HER) at the cathode and the oxygen evolution reaction (OER) at the anode. The HER includes two electron-transfer steps [6] while the OER is more sluggish with an energetically uphill four-electron transfer [7,8]. During the water-splitting process, ionized hydrogen ions are equivalent to the hydroxide ions. Therefore, hydrogen and oxygen can be produced through PEC overall water splitting at the same time without adding a sacrificial agent [9,10]. However, it is difficult to search for spontaneous and efficient catalysts for overall water splitting because they should not only satisfy the stringent requirements for PEC water splitting but also show efficient HER and OER behaviors [11,12,13,14].

Several bifunctional catalysts have been designed and, though some of them can release holes and excite electrons to oxidize water and reduce photons, their catalytic activity is poor [15,16,17]. Additionally, some exhibit good behavior in a single reaction while improving the performance in the other reaction [18,19,20]. Developing promising and efficient semiconductor bifunctional photocatalysts remains a great challenge. Recently, monoclinic clinobisvanite bismuth scheelite (ms-BiVO_4_) has received extensive concern due to the ideal gap (2.4 eV) for visible-light absorption, suitable valence band edge position (~2.8 V vs. RHE), and superior hole mobility [21,22]. As we know, BiVO_4_ single crystals with a different fraction of exposed facets have been synthesized, exhibiting different PEC activity [23,24,25]. In this case, the (110) facet is highly exposed with good stability and is easy to separate and accumulate photogenerated charges. Nevertheless, the PEC performance of undecorated BiVO_4_ is mediocre due to its shortcomings in fast surface electron-hole recombination and sluggish water redox behaviors [26,27]. Various improvement strategies, such as combining heterostructure catalysts, doping with impurities, and so on [28,29,30,31,32], have been adopted to overcome these shortcomings. In this case, oxygen vacancy modification had a beneficial influence on water oxidation in BiVO_4_ owing to creating more active sites and favoring charge separation and transfer [33,34,35,36]. There are many ways to induce and increase oxygen vacancies in BiVO_4_. For example, Xu et al. applied an ionized argon plasma technology on three-dimensional nanoporous BiVO_4_ to controllably generate surface oxygen vacancies [23]. Mayur et al. demonstrated a rapid Fenton-like reaction method for fabricating an ultrathin amorphous Ni:FeOOH nanolayer with an in-situ-induced O_vac_ on the BiVO_4_ photoanode [24].

In previous work, BiVO_4_ (110) facet modified by oxygen vacancies has been investigated to meet the requirements of PEC water splitting [37], which have been verified by experiments [23,24,25]. Under irradiation, photogenerated electrons and holes can provide driving forces for the HER and OER. Though some works have investigated the effect of the O_vac_ on the BiVO_4_, they mainly discussed the stability and electronic structures. In this work, we focus on the effect of the O_vac_ on bifunctional PEC activity, simultaneously, for the HER and OER in the BiVO_4_ (110) facet, not only analyzing the electric structure but also describing the thermodynamic process for the HER and OER. The results show that oxygen vacancy not only decreases the hydrogen adsorption free energy (Δ*G*_H*_) of the HER but also reduces the overpotential of the OER; this is mainly because the O_vac_ induces charge rearrangement to enhance charge transfer from the active site to adsorbate atoms. Interestingly, during the water-splitting process, under the interference of hydrogen and oxygen, more photogenerated charges can be excited. Additionally, the O_vac_ reduces the work function and favors the carrier’s transition from the inner to the surface. As a result, the introduction of the O_vac_ can effectively improve bifunctional PEC activity in the BiVO_4_ (110) facet.

## 2. Computational Model and Methods

To investigate the HER and OER performance of the BiVO_4_ (110) facet, the Vienna ab initio simulation package (VASP) was employed by the density functional theory (DFT). The generalized gradient approximation (GGA) was used with Perdew–Burke–Ernzerhof (PBE) [38,39,40,41], the energy cutoff was 400 eV, the convergent criterion was an energy change less than 1.0 × 10^−5^ eV, and the optimization was completed when the maximum forces were 0.01 eV·Å^−1^. The *k*-points were set to 5 × 5 × 1 for geometric optimization and 7 × 7 × 1 for electronic structure calculations. We compared the GGA-calculated lattice parameters to HSE methods. The GGA-calculated lattice parameters *a* = 5.04 Å, *b* = 5.27 Å, and *c* = 11.89 Å were well matched with the experimental values (*a* = 5.10 Å, *b* = 5.17 Å, and *c* = 11.69 Å) [42,43]. Compared to the HSE06-calculated band gap of 2.80 eV [44], the band gap obtained by the GGA method of 2.25 eV was closer to the experimental value of 2.45 eV [43], proving the reliability of the GGA method. The bilayer (110) facet was cleaved from the optimized ms-BiVO_4_ bulk and a 12 Å vacuum slab was added along the z-direction to minimize the potential artificial interaction between the adjacent images, as shown in Figure 1a. To investigate the effect of the O_vac_ on the active atom, we built the O_vac_ neighboring to the active Bi atom, as shown in Figure 1b.

## 3. Results and Discussion

### 3.1. The Effect of the O_vac_ on the Electronic Structure of the BiVO_4_ (110) Facet

In our previous work, we discussed the effect of the O_vac_ on BiVO_4_ in detail. The formation energy can be defined as *E*_form_ = *E*_Ovac_ − *E*_surf_ + 1/2 *E*_O2_, where *E*_Ovac_, *E*_surf_, and *E*_O2_ are the total energies of BiVO_4_ (110) facets with and without the O_vac_ and molecular O_2_, respectively. The formation energy is 3.86 eV and the calculated result is comparable to the previous report, indicating oxygen vacancies can be easily formed in the BiVO_4_ (110) facet [37]. Additionally, the O_vac_ induces some changes in the structure. It is these changes that result in the change in electrical structure. As shown in Figure 1, BiVO_4_ (110) facets without and with the O_vac_ are all semiconductors with band gaps of 2.28 eV and 2.33 eV. Similarly, the VBM mainly consists of V 3d and O 2p and the CBM is primarily composed of Bi 6p, O 2p, and V 3d. Differently, owing to the presence of the O_vac_, electron density redistributes and localized states appear in the band gap of BiVO_4_, whose components are V 3d and O 2p, promoting charge carrier concentration and mobility [37]. The work function decreases from 6.04 eV to 5.87 eV, the carrier’s transition from the solid to the vacuum region becomes easy, and, therefore, more photogenerated carriers will participate in catalytic reactions. More importantly, owing to the O_vac_-induced internal electric field, the CBM upshifts and straddles the water reduction potential. As a result, the BiVO_4_ (110) facet satisfies the requirements of PEC water splitting by introducing the O_vac_; the detailed analysis of electronic structure, optical adsorption, and band edge alignment can be found in Ref. [37]. Photogenerated electrons and holes can provide the charges for further HERs and OERs.

### 3.2. The Effect of the O_vac_ on the HER of the BiVO_4_ (110) Facet

As we know, the acidity and alkalinity of the solution will affect catalytic performance. Compared to acid HER behavior, the alkaline HER is more complex. Here, we focus on discussing the effect of the O_vac_ on active sites; thus, PEC water splitting in acidic conditions is considered. During the HER process, hydrogen adsorption on the BiVO_4_ surface takes place; the adsorption-free energy (Δ*G*_H*_) is an important descriptor for evaluating catalytic activity, which can be defined as Δ*G*_H*_ = *E*_H_ + Δ*E*_ZPE_ − *T*Δ*S*_H_, where *E*_H_ is hydrogen adsorption energy, Δ*E*_ZPE_ is the difference in zero-point energy of hydrogen vibration between the adsorbed state and the gas phase, and Δ*S*_H_ is the entropy difference between the adsorbed state of the system and gas phase at the standard condition [45]. The ideal Δ*G*_H*_ is nearly zero because the weak binding (Δ*G*_H*_ > 0) reduces the sustainability of the reaction and the strong binding (Δ*G*_H*_ < 0) makes hydrogen difficult to capture from the catalysts [46]. In the BiVO_4_ (110) facet, the surface O atom is the active site for the HER and surface different O atoms, respectively, act as active sites for H adsorption in Figure 1. The Δ*G*_H*_ values are −0.58, −0.49, and −0.56 eV for O1, O2, and O3, respectively; the more negative values indicate stronger binding between the active site and H* but strong binding makes it difficult for hydrogen ions to separate from the surface; thus, O2 is considered a HER active site for further investigation. 

As shown in Figure 2, with the assistance of the O_vac_, the Δ*G*_H*_ of the active O2 site decreases to −0.39 eV, indicating the O_vac_ weakens the binding between the H adatom and surface, and hydrogen dissociation becomes easy. This is because the appearance of the O_vac_ induces unsaturated bonds and unsaturated electrons, which leads to charge rearrangement; as a result, there is 7.19 e on the active site of O2 on the (110)-O_vac_ surface, slightly more than that on the (110) surface with 7.16 e based on Bader charge analysis. Compared to Figure 2a, Figure 2b shows that shallow energy levels appear near the VBM, mainly deriving from V and O atoms; this shallow level should be the donor level introduced by the n-doping, as shown by a peak above the Fermi level. Importantly, owing to H intervention, more states appear at intermediate levels in the H-adsorbed (110)-O_vac_ facet than the (110)-O_vac_ facet, meaning H not only participates in the HER but also promotes the generation of photogenerated charges, thus benefiting optical adsorption enhancement.

### 3.3. The Effect of the O_vac_ on the OER of the BiVO_4_ (110) Facet

For the OER, there are four electron-transfer steps, more complicated than the HER, and each step involves electron transfer accompanied by proton removal:(1)$$H_{2}O\;(l)+\;*\to \mathrm{HO}*+{\;H}^{+}+{\;e}^{-},$$
(2)$$\mathrm{HO}*\to O*+{\;H}^{+}+{\;e}^{-},$$
(3)$$O*+{\;H}_{2}O\to \mathrm{HOO}*+{\;H}^{+}+{\;e}^{-},$$
(4)$$\mathrm{HOO}*\to O_{2}\left(g\right)+{\;H}^{+}+{\;e}^{-},$$
where * represents the surface adsorption site.

As we know, the active site of BiVO_4_ for the OER is the surface Bi atom; four different Bi atoms of the surface are considered H_2_O* adsorption sites and the adsorption energies are defined as *E*_ads_ = *E*_tot_ − *E*_surf_ − *E*(*) (where *E*_tot_, *E*_surf_, and *E*(*) are total energies of (110) facets with and without adsorbate and the energy of adsorbate), which are −0.60, −0.36, −0.46, and −0.67 eV for Bi1, Bi2, Bi3, and Bi4 atoms, respectively. All are exothermic and the binding strength between the Bi4 atom and surface is more energetically favorable; thus, Bi4 is considered the active site of the OER for further investigation. During the first reaction step, H_2_O* is adsorbed on top of the Bi atom; the Bi–O bond length is 2.57 Å and the H_2_O* adsorption energy is −0.67 eV in the (110) facet. For the (110)-O_vac_ facet, the bond length of Bi–O decreases to 2.44 Å and the *E*_ads_ of H_2_O* decreases to −0.82 eV. The changes come from the O_vac_-induced charge redistribution; a more unsaturated surface is more hydrophilic, displaying stronger interaction between H_2_O* and the surface [47]. Compared to Figure 3a, more impurity levels appear at the VBM and CBM in Figure 3b, reducing the band gap; additionally, the impurity levels also appear at the band gap for O* and HOO* adsorbed on BiVO_4_ (110). Additionally, the PDOS of H_2_O* adsorption in Figure 3 shows great changes have taken place not only on Bi atoms but also on V atoms in the (110)-O_vac_ facet. The charge density difference (see inset in Figure 3) further displays great charge transfer in the (110)-O_vac_ facet occurs not only on the surface Bi atom but also on the neighboring V atom, indicating the adjacent atoms have a great effect on the catalytic reaction owing to the existence of the O_vac_. Additionally, the isolated states on the band gap move to the conduction band when the (110)-O_vac_ facet adsorbs H_2_O*, indicating that H_2_O is easily adsorbed on the surface owing to the existence of the O_vac_. Similar to H_2_O* adsorption, the bond lengths of Bi–O decrease greatly to 2.109, 2.164, and 2.193 Å for HO*, O^*,^ and HOO* adsorption in the (110)-O_vac_ facet while they are 2.265, 2.280, and 2.636 Å in the pure (110) facet. The shorter bond lengths indicate stronger interaction between the surface Bi atom and O atom and the charge transfer from surface Bi to the adsorbed O atom becomes more easy. 

In addition, the O_vac_-induced states are significantly enhanced for the HO*, O*, and HOO* adsorbed in the (110)-O_vac_ facet, indicating more photogenerated charges can be excited in the OER process, which can be verified by work functions in Figure 4. The work functions are 6.01, 6.63, 6.25, and 5.75 eV for H_2_O*, HO*, O*, and HOO* adsorption in the pure (110) facet; they reduce greatly to 4.05, 4.32, 4.77, and 4.82 eV in the (110)-O_vac_ facet. The reduced work functions show the charges are more easily transferred from bulk to surface in each electron step; simultaneously, the charges possess higher energies and stronger oxidation capacity, agreeing with experimental results that the O_vac_ favors charge separation and transfer [23].

To gain insight into the thermodynamics and direction of the reaction, we calculate the free energy diagrams of the OER process on BiVO_4_ (110) facets. The free energy can be defined as Δ*G* = Δ*E* + ΔZPE-*T*Δ*S*, in which Δ*E* is the reaction energy depending on each electron-transfer step, ΔZPE denotes the change of zero point energies, and Δ*S* is the change in entropic contribution by employing the computed vibrational frequencies and standard tables for the reactants and products in the gas phase [48,49]. In the free energy calculation, an external bias *U* can be applied on each electron-transfer step. The free energies can be expressed as:(5)$$\Delta G_{1}=E\left(\mathrm{HO}*\right)-E_{\mathrm{surf}}-E_{H_{2}O}+\frac{1}{2}E_{H_{2}}+(\Delta \mathrm{ZPE}-T\Delta S)-eU$$
(6)$$\Delta G_{2}=E\left(O*\right)-E\left(\mathrm{HO}*\right)+\frac{1}{2}E_{H_{2}}+(\Delta \mathrm{ZPE}-T\Delta S)-eU$$
(7)$$\Delta G_{3}=E\left(\mathrm{HOO}*\right)-E\left(O*\right)-E_{H_{2}O}+\frac{1}{2}E_{H_{2}}+(\Delta \mathrm{ZPE}-T\Delta S)-eU$$
(8)$$\Delta G_{4}=E_{\mathrm{surf}}-E\left(\mathrm{HOO}*\right)+E_{O_{2}}+\frac{1}{2}E_{H_{2}}+(\Delta \mathrm{ZPE}-T\Delta S)-eU$$

The total reaction $H_{2}O\to \frac{1}{2}O_{2}+{\;H}_{2}$ occurs at the potential of 2.46 V per water molecule; the minimum free energy should split two H_2_O molecular with 4.92 V [50].

Figure 5a shows the free energies of the OER process in the BiVO_4_ (110) facet at U = 0 V, pH = 0, and T = 298 K; the first step is the adsorption of H_2_O* moiety on the surface Bi site and removing a proton from H_2_O* to create an HO* radical and the free energy runs steeply uphill to 2.66 eV. This process consumes more energy than other processes, which is a rate limitation, indicating water dissociation is difficult. The reaction continues and the HO* moiety releases another proton generating O*; the free energy is 3.86 eV. The O* is very electrophilic and immediately gains an electron by bonding with adjacent H_2_O* and forming HOO* with the free energy of 5.34 eV. Lastly, O_2_ is released from the (110) surface. The free energies of the four processes are all uphill; it is necessary to apply potential bias (or overpotential) to make each step downhill. The overpotential can be obtained by the difference between the voltage for all free-energy steps downhill and the minimum voltage required for the OER [51]. For the (110) facet, the first step to form the HO* radical is very critical to determine the limiting potential with the overpotential of 1.43 V (=2.66 − 1.23 V). Importantly, the overpotential of the (110)-O_vac_ facet decreases greatly with 0.62 V and the third step for generating the HOO* radical becomes the determining-potential step (see Figure 5b).

The reduction in overpotential originates from charge rearrangement, which can be found from the Bader charge on the active site Bi at each electron step in Figure 6. For the pure BiVO_4_ (110) facet, the active site Bi atom has the most charges at the first step with 3.22 e, displaying weak oxidation properties, which limits the reaction; thus, the first step is the rate-determining step. Meanwhile for the (110)-O_vac_ facet, the charges on the Bi atom overall decrease, indicating greater charge transfer from active Bi to neighboring atoms. Additionally, the Bi atom has the most charges with 3.10 e at the third step, which becomes the rate-control step. Obviously, the presence of the O_vac_ induces charge rearrangement and accelerates charge transfer then effectively enhances the activity of oxygen generation.

Furthermore, we investigate the effect of the O_vac_ position and the O_vac_ concentration on the OER. As shown in Figure 7a, we create the O_vac_ coordinated with V, comparing with the O_vac_ coordinated with Bi (see Figure 5b); the second step becomes the rate-control step with the overpotential of 0.71 V. Additionally, we create two and three O_vacs_ to increase the O_vac_ concentration to 6.25% and 9.38%, as shown in Figure 7b,c. The rate control steps occur at the fourth step with the overpotential of 0.32 V for a 6.25% O_vac_ concentration and the third step with the overpotential of 0.63 V for a 9.38% O_vac_ concentration. These indicate that the O_vac_ site and concentration have a significant influence on OER properties.

## 4. Conclusions

Based on the DFT calculation, we investigate the effect of the O_vac_ on the catalytic activity of overall water splitting in the BiVO_4_ (110) facet. The results have shown that the O_vac_ induces charge rearrangement; there appear intermediate shallow levels, favoring charge transfer from the VBM to the CBM. During the HER and OER processes, the intervention of H and O promotes the generation of photogenerated charges. Meanwhile, there occurs great charge transfer between the active site and neighboring atoms, which not only simulates the neighboring atom activity but also enhances the interaction between the active sites. Additionally, the O_vac_ reduces the work function, which not only benefits theoverflow of charge but also enhances charge oxidation capacity. Therefore, the (110)-O_vac_ facet has lower Δ*G*_H*_ for the HER and the overpotential for the OER. Contributing from the O_vac_, bifunctional catalytic activity can be effectively improved in the BiVO_4_ (110) facet. Importantly, vacancy-defect engineering is a feasible strategy to improve the PEC water-splitting activity of BiVO_4_.

## Figures and Tables

**Figure 1 nanomaterials-14-01270-f001:**
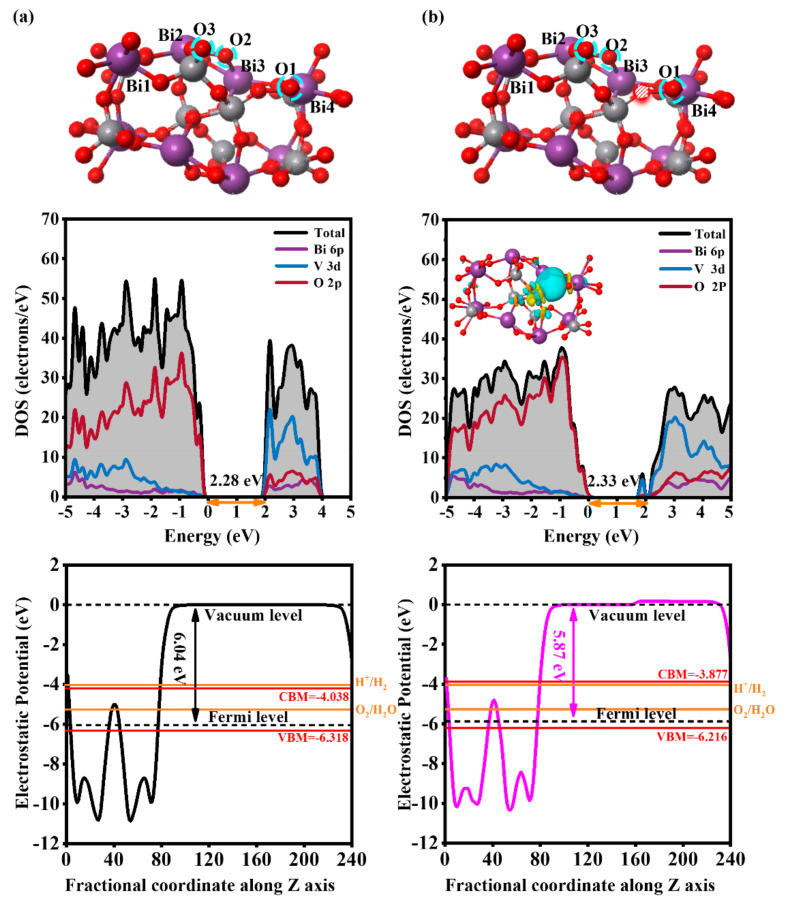
Optimized structures, the density of states (DOS), and the work functions compared to the redox potential of water in BiVO_4_ (110) facets (**a**) without and (**b**) with the O_vac_.

**Figure 2 nanomaterials-14-01270-f002:**
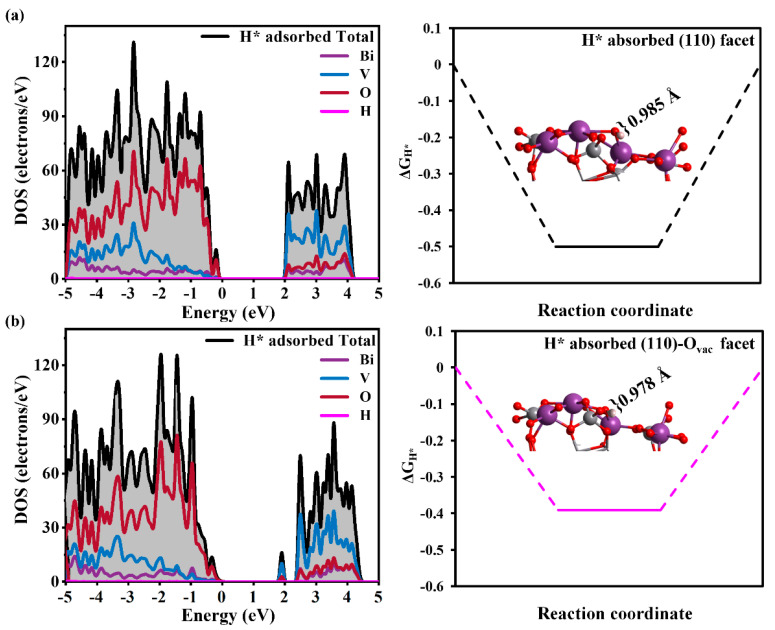
The total and partial DOS of H adsorption and hydrogen adsorption free energy (Δ*G*_H*_) in BiVO_4_ (110) facets (**a**) without and (**b**) with the O_vac_.

**Figure 3 nanomaterials-14-01270-f003:**
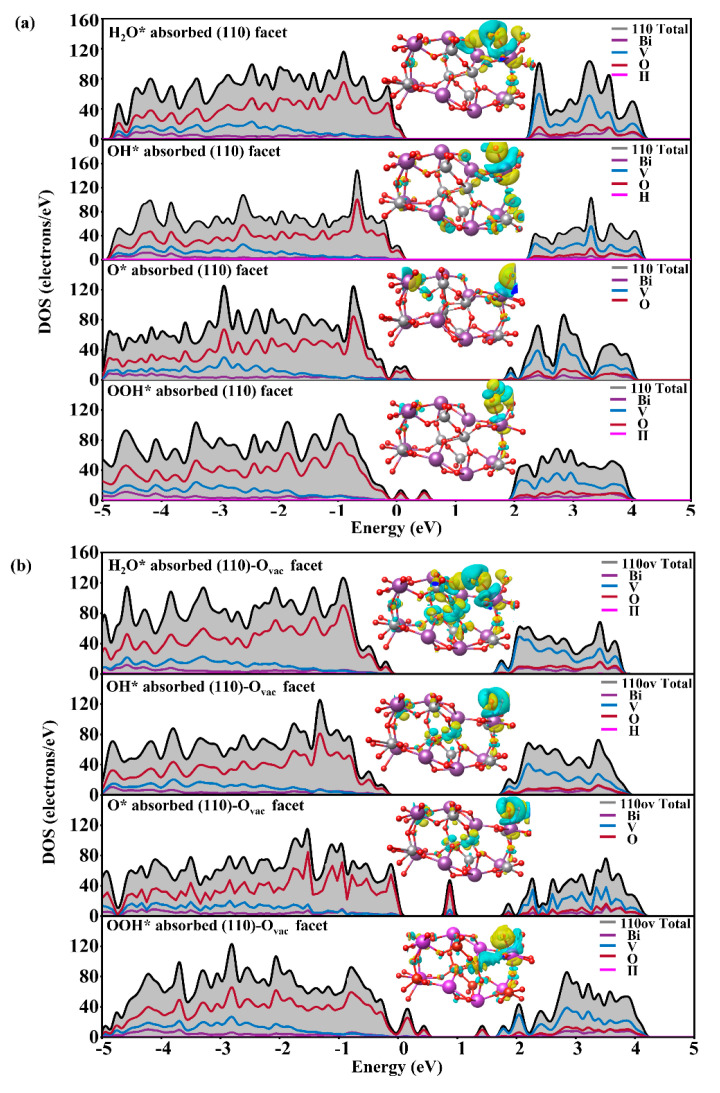
DOS of H_2_O* HO*, O*, and HOO* adsorbed on BiVO_4_ (110) facets (**a**) without and (**b**) with the O_vac_. The insets are charge density difference and the yellow and blue colors represent charge accumulation in depletion.

**Figure 4 nanomaterials-14-01270-f004:**
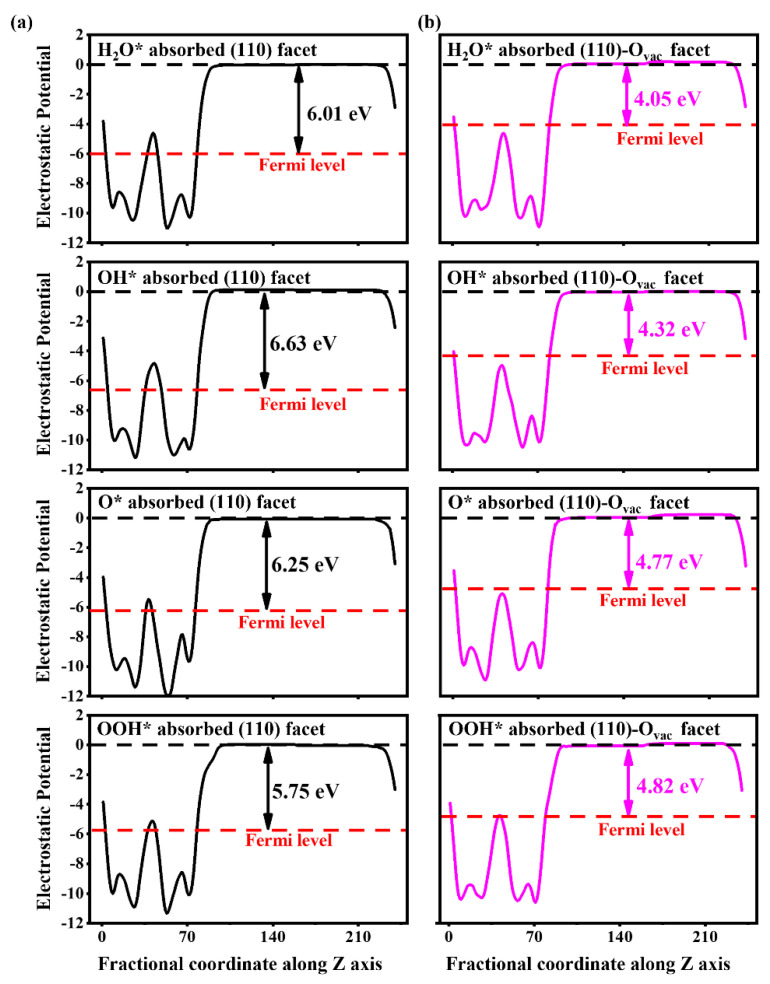
Work functions of H_2_O*, HO*, O*, and HOO* adsorbed on BiVO_4_ (110) facets (**a**) without and (**b**) with the O_vac_.

**Figure 5 nanomaterials-14-01270-f005:**
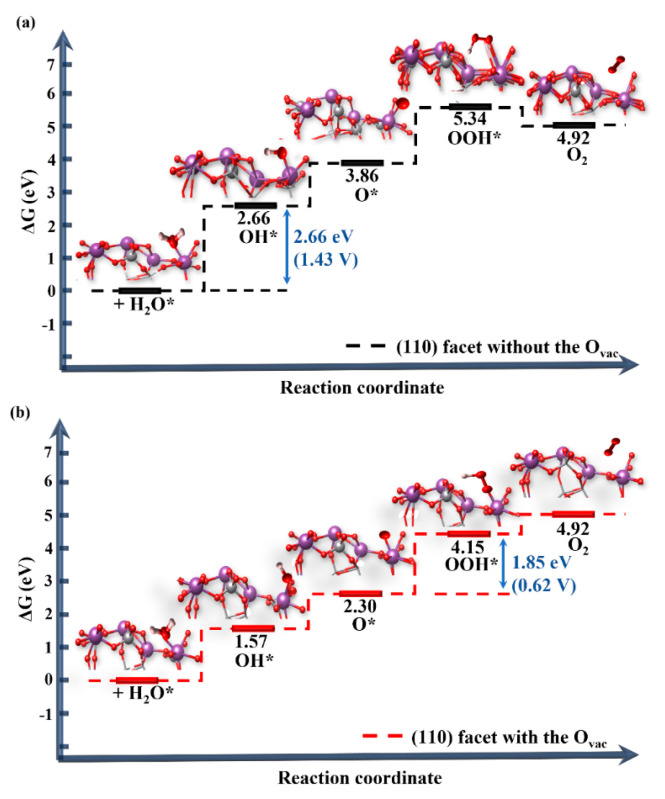
Free energy diagrams for the four steps of the OER on the BiVO_4_ (110) facets without (**a**) and with (**b**) the O_vac_ at *U* = 0 eV, pH = 0, and *T* = 298 K.

**Figure 6 nanomaterials-14-01270-f006:**
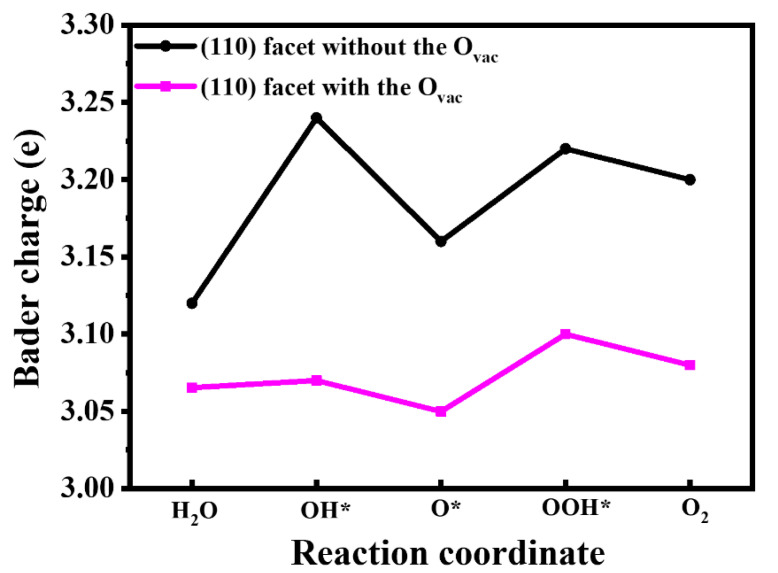
The Bader charge of active site Bi at each electron step for the OER in the BiVO_4_ (110) facets with and without the O_vac_.

**Figure 7 nanomaterials-14-01270-f007:**
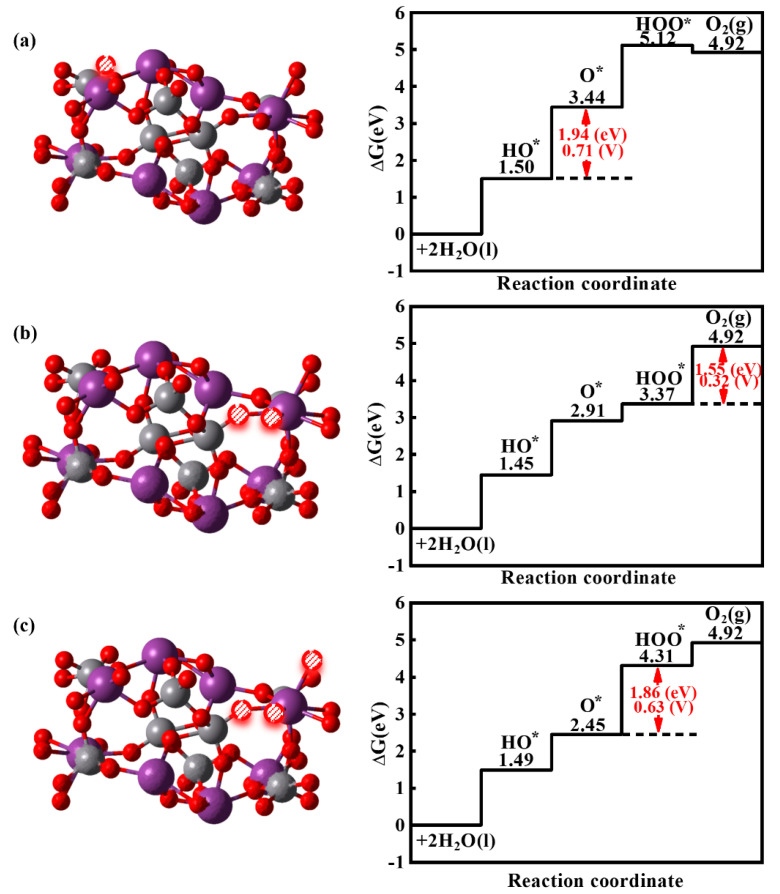
Optimized structure and free energy diagrams for the four steps of the OER on the BiVO_4_ (110) facets with different O_vac_ sites and concentrations of (**a**) 3.13%, (**b**) 6.25%, and (**c**) 9.38%.

## Data Availability

The data that support the findings of this study are available within the article.

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
