# Peer review of "Oxygen-Vacancy-Induced Enhancement of BiVO4 Bifunctional Photoelectrochemical Activity for Overall Water Splitting"

_nanomaterials, 2024, doi:10.3390/nano14151270_

Round 1

Reviewer 1 Report

Comments and Suggestions for Authors

This work is devoted to studying the influence of Ovac c on the catalytic activity of overall-water-splitting in BiVO4 (110) facet. The work is theoretical and based on calculations using the DFT method. This work is a continuation of the authors’ previous works in this direction. The authors presented competent and convincing calculations, including calculations of the free energies of the reactions.

Of course, the work is of some interest for the development of bifunctional catalysts, but there are some comments.

1) In the introduction, the authors should have described more clearly and in detail the real ways of increasing oxygen vacancies in BiVO4.

2) Is there experimental confirmation of the presented calculations? In a previous study, the authors concluded that the BiVO4 [110] facet could become a promising photocatalyst for water splitting. Considering that the authors’ previous similar work was published in 2021, there was enough time to test the calculations in practice.

Author Response

Comment1:This work is devoted to studying the influence of Ovac c on the catalytic activity of overall-water-splitting in BiVO4 (110) facet. The work is theoretical and based on calculations using the DFT method. This work is a continuation of the authors’ previous works in this direction. The authors presented competent and convincing calculations, including calculations of the free energies of the reactions.

Of course, the work is of some interest for the development of bifunctional catalysts, but there are some comments.

Response: We sincerely thank the Reviewer for the valuable evaluation and support on our work. We also appreciate the constructive suggestions, which can enhance the quality of our manuscript. Based on Reviewer’s kind suggestions, we have added  corresponding analyses, and the presentation of the manuscript has also been carefully amended. Our point-by-point responses are presented below.

1) In the introduction, the authors should have described more clearly and in detail the real ways of increasing oxygen vacancies in BiVO4.

Reply: Thank the referee for the comments. There are many ways to induce and increase oxygen vacancies in BiVO4. For example, Xu et al applied an ionized argon plasma technology on three-dimensial nanoporous BiVO4 to controllable generate surface oxygen vacancies [Applied Catalysis B: Environmental 2021, 281, 119477]. Mayur et al demonstrated a rapid Fenton-like reaction method for fabricating an ultrathin amorphous Ni:FeOOH nanolayer with in situ-induced Ovac on the BiVO4 photoanode [ACS Appl. Mater. Interfaces 2023, 15, 21123]. We have added related discussion on part of Introduction in the revised manuscript.

2)Is there experimental confirmation of the presented calculations? In a previous study, the authors concluded that the BiVO4 [110] facet could become a promising photocatalyst for water splitting. Considering that the authors’ previous similar work was published in 2021, there was enough time to test the calculations in practice.

Reply: Thank you for your valuable feedback and suggestions. There are experimental works have been verified the conclusion of our previous work published in 2021. For example, Mayur et al found the existence of the oxygen vacancies enhanced water oxidation activity and stability of BiVO4 photoanodes, because the vacancies improved charge separation and charge transfer efficiency [ACS Appl. Mater. Interfaces 2023, 15, 21123]. Xu et al found oxygen vacancies activated surface reactivity to favor charge separation and transfer and therefore improved the photocatalytic performance in BiVO4 [Applied Catalysis B: Environmental 2021, 281, 119477].   

Comment 2:I have reviewed the manuscript titled "Oxygen vacancy-induced enhancement of BiVO4 bifunctional photoelectrochemical activity for overall water splitting," which is well-written and effectively supports its main thesis. However, before its acceptance in Nanomaterials, there are some minor issues that should be addressed. 

Response: We sincerely thank Reviewer-2 for the constructive suggestions, which are greatly valuable to improve the quality of this work. we have attached a point-to-point reply to your concerns below and hope indeed publishable on the journal.

Specifically, the authors should clarify the differentiation between their work and existing literature, particularly in

Chem. Mater. 2018, 30, 21, 7793–7802

Structural and electronic properties of oxygen defective and Se-doped p-type BiVO4 (001) thin film for the applications of photocatalysis, Applied Catalysis B: Environmental, Volume 224, 2018, Pages 895-903

10.1039/D2RA04890F (Paper) RSC Adv., 2022, 12, 31317-31325.

It would strengthen the introduction to include a comparative discussion on these references. Furthermore, the manuscript should elaborate on how the stability of oxygen vacancies (Ov) in BiVO4 is determined. Lastly, the significance of Density of States (DOS) in optimizing Oxygen Evolution Reaction (OER) and Hydrogen Evolution Reaction (HER) catalysts should be explicitly stated within the text for better clarity and understanding. These revisions will enhance the manuscript's coherence and academic rigor.

 Reply: Thank you for your valuable feedback and suggestions. We respectively answer these three questions.

Question 1: It would strengthen the introduction to include a comparative discussion on these references.

Reply: In Chem. Mater. 2018, 30, 21, 7793: Hosung et al investigated the stability and the conductivity of oxygen vacancy (Ovac) and N-doped BiVO4. The results showed the Ovac and substitution nitrogen affected the formation energy of polarons, effectively contributing to an increase of the carrier mobility. In Applied Catalysis B: Environmental, Volume 224, 2018, 895: Habib et al investigated the structural and electronic properties of the Ovac and Se doped BiVO4 (001) facet. The calculated results have shown that Se-doped BiVO4 displayed p-type semiconductor behavior, both Ovac and Se doped BiVO4 had ideal band edge, band gap and small effective mass of electrons and holes. In RSC Adv., 2022, 12, 31317: Zhang et al investigated the electronic structure of BiVO4 (001) and (011) facets and found the Ovac made the BiVO4 become an n-type semiconductor, Bi and V vacancies made the BiVO4 became an n-type semiconductor. In our work, we investigated the effect of oxygen vacancy on the photoelectrochemical activity of the BiVO4 not only analyzed the electronic structure including band structure and work function but also described thermodynamic process for HER and OER. We have added related discussions in page 2 in the revised manuscript.

Question 2: Furthermore, the manuscript should elaborate on how the stability of oxygen vacancies (Ov) in BiVO4 is determined.

Reply: The stability of BiVO4 with the oxygen vacancy can be investigated by calculating the formation energy as Eform = EOvac - Esurf +1/2 EO2, where EOvac, Esurf and EO2 are the total energies of BiVO4 [110] facets with and without the Ovac and molecular O2, respectively. The formation energy of (110) facet with the Ovac is 3.86 eV, the calculated result is comparable to those reported in Reference [RSC Advances, 2017, 7, 9130], indicating oxygen vacancies can be easily formed in the BiVO4 (110) facet. We have added the stability calculation and discussion on page 1 in the revised manuscript. 

Question 3: Lastly, the significance of Density of States (DOS) in optimizing Oxygen Evolution Reaction (OER) and Hydrogen Evolution Reaction (HER) catalysts should be explicitly stated within the text for better clarity and understanding.

Reply:Compared to Fig. 2(a), Fig. 2(b) shows there appear shallow energy levels near the VBM mainly deriving from V and O atoms, this shallow level should be the donor level introduced by the n-doping, a shown by a peak above the Fermi level. Compared to Fig. 3(a), there are more impurity levels appears at the VBM and CBM in Fig. 3(b), reducing the band gap, additionally, there also appear the impurity levels at the band gap for O*, and HOO* adsorbed on BiVO4 (110). We have added these description on pages 3-4 in the revised manuscript. 

Reviewer 2 Report

Comments and Suggestions for Authors

I have reviewed the manuscript titled "Oxygen vacancy-induced enhancement of BiVO4 bifunctional photoelectrochemical activity for overall water splitting," which is well-written and effectively supports its main thesis. However, before its acceptance in Nanomaterials, there are some minor issues that should be addressed.

Specifically, the authors should clarify the differentiation between their work and existing literature, particularly in

Chem. Mater. 2018, 30, 21, 7793–7802

Structural and electronic properties of oxygen defective and Se-doped p-type BiVO4(001) thin film for the applications of photocatalysis, Applied Catalysis B: Environmental, Volume 224, 2018, Pages 895-903

10.1039/D2RA04890F (Paper) RSC Adv., 2022, 12, 31317-31325.

It would strengthen the introduction to include a comparative discussion on these references. Furthermore, the manuscript should elaborate on how the stability of oxygen vacancies (Ov) in BiVO4 is determined. Lastly, the significance of Density of States (DOS) in optimizing Oxygen Evolution Reaction (OER) and Hydrogen Evolution Reaction (HER) catalysts should be explicitly stated within the text for better clarity and understanding. These revisions will enhance the manuscript's coherence and academic rigor.

Author Response

(The authors gave the same response as above.)
